# Self-Determined Regulation, Achievement Goals and Sport Commitment in Spanish Masters Swimmers

**DOI:** 10.3390/bs13100828

**Published:** 2023-10-09

**Authors:** Juan Ángel Simón-Piqueras, Pedro Gil-Madrona, David Zamorano-García, Miriam De La Torre-Maroto, Juan Gregorio Fernández-Bustos

**Affiliations:** Faculty of Education, Castilla-La Mancha University, 02071 Albacete, Spain; pedro.gil@uclm.es (P.G.-M.); miriamla.torre@alu.uclm.es (M.D.L.T.-M.); juang.fernandez@uclm.es (J.G.F.-B.)

**Keywords:** Masters swimmers, sport masters, motivation, mastery, wellness, social relations in masters

## Abstract

This work analyses the motivational regulation, achievement goals and sport commitment of Spanish Masters swimmers, being the first study of its kind. A total of 211 (106 women and 105 men) Masters swimmers from the Federation of Castilla-La Mancha (Spain) aged between 20 and 74 participated. Each participant completed the Sport Motivation Scale (motivation regulation), the Achievement Goals Questionnaire for Sport (achievement goals) and the Sport Commitment Questionnaire (sport commitment), all validated for the Spanish population. A mixed rANOVA was performed to analyse the results, using sex and age group as covariates, and the different groupings of the sample (weekly training days, weekly training hours and practice of other sports) as between-participants factors. The results showed that the participants presented a predominantly intrinsic–integrated and identified regulation, rather than introjected and external regulation. Mastery goals were more prevalent than performance–approach and performance–avoidance goals. In addition, current sport commitment was high, being greater than future commitment. Pearson’s correlation analysis showed moderate relationships between our variables. We found no influence of either the covariates of sex and age or the grouping variables. These findings serve as the basis for further study of the motivation of Masters swimmers in Spain.

## 1. Introduction

### 1.1. The Masters Swimming Phenomenon

Since the late 1970s and early 1980s, a number of national swimming federations have promoted competitive activity in the different specialties they are responsible for (swimming, artistic swimming, diving, water polo and open water swimming). These activities are aimed at athletes aged from 20–30 years (depending on the criteria of the federation and sport specialty) to practically unlimited ages. Such athletes are known as Masters swimmers (hereon in masters). In recent decades, World Aquatics itself (formerly known as FINA, the International Swimming Federation) has fostered specialties in Masters athletes through organizing world championships. The participation of masters in such events has increased exponentially [1], with approximately 10,000 athletes from more than 100 countries participating in the last world championships. It is thus a truly global phenomenon [2], which World Aquatics relates to the promotion of physical activity and health, friendship, understanding and competition between adult athletes, with no age limit imposed on participation in the programme [2].

In the context of Spain, the Royal Spanish Swimming Federation (RFEN, in its Spanish acronym) and the corresponding federations of Spain’s autonomous communities have replicated the organization of similar events on their own scale, driven by the increased number of participants in such events. According to data from the RFEN, the number of masters has risen by 550% between 2010 and 2023, decreasing minimally only during the period of the COVID-19 pandemic [3]. A closer look at the information channels of these institutions reveals an intense programme of sporting events, with characteristics that encourage all participants to find their motivation. For example, the age categories are grouped into five-year age ranges, such that, if the effect of age impacts performance, when changing category, the masters will again have the advantage of being in the youngest year in their group. Another element of interest we find is that, in many cases, the nature of the competition means it is more important to “contribute points” to one’s team than the individual result, thus transforming a clearly individual sport into a team one [4,5]. In addition, there is a standardized scoring system according to the age group to which the participant belongs, meaning that it can be ascertained through these points whether they improve with age even though the scores are lower. All these strategies are based on scientific contributions [6,7]. However, we have also observed that these institutions limit themselves to organizing events and, at least in those reviewed, we found no intervention programmes other than those containing superficial information [8].

This leads us to the question of whether these events are organized to promote the activity, or whether they are the consequence of the demand resulting from the increase in the number of masters. Can competition alone favour practice, or are other factors linked to personal, salutogenic, motivational and personal well-being aspects arguably involved, as suggested by studies on engagement in physical activity and sport [8,9,10,11]?

In order to answer the above question, a first approach to resolving this issue might lie in determining the profiles of the Masters swimmers analysed in terms of their predominant motivation regulations, their achievement goals and their current and future commitment to sport. In this sense, it would be interesting to correlate these aspects with each other.

### 1.2. Self-Determined Regulation Theory

Self-determination theory (SDT) is currently the predominant theory of human motivation [12]. SDT explains how human motivation can be affected by personal, social and environmental factors [13,14]. It is composed of different sub-theories, which, due to the complexity of the proposed framework, help examine different aspects of motivation. One of the main sub-theories is known as Organismic integration theory [12], which addresses the continuum of self-determination, and suggests that the degree of self-determination involved in behaviours depends on the social factors that hinder the integration of behavioural regulation [15]. This sub-theory posits six types of motivation, according to the degree of self-determination. One of these types is intrinsic regulation, which is related to the satisfaction involved in performing an activity. Another type is extrinsic motivation, which is divided into integrated regulation, which links an activity to lifestyle, values and goals; identified regulation, which is associated with the benefits of the activity; introjected regulation, characterized by feelings of guilt or pride; and external regulation, which involves rewards or prizes. The final type is amotivation, which involves a participant finding no sense in continuing to engage in an activity [14].

SDT provides a broad theoretical framework to understand the conditions that promote high motivation in sport [16]. Broadly speaking, research has fundamentally focused on aspects related to the differentiation between intrinsic and extrinsic regulations. As for sports in general, the results are mixed. Some studies show the prevalence of external regulation, which helps explain that competition and its rewards alone can foster an increase in practice [17], while others report the key role of more self-determined types of regulation, suggesting there are more significant factors than competitive prizes [18]. Meanwhile, other approaches point to regulation being an underlying construct that is present to varying degrees in certain moments of the pursuit of sport [19,20]. Regarding the question of what type of motivation is predominant in sport, and specifically among Masters athletes, the findings are disparate [17,19,20], and, to our knowledge, a motivational regulation profile has not yet been presented in any study.

Regarding Masters athletes, we found no studies that provide this profile. The few studies on motivation on such Masters have essentially focused on analysing variables that may be associated with more self-determined [21] or less self-determined regulation [17]. Hence, we consider it of great importance to establish a detailed profile based on SDT.

### 1.3. Achievement Goal Theory

Another theory widely used to explain motivation in sports is that known as achievement goal theory (AGT). Achievement goals are defined as the reasons or aims that drive an individual’s behaviour [22]. This theory holds that the aim of an individual in achievement situations is to show ability, and that, depending on what such individuals consider achievement, they will present different types of goals [23,24]. AGT has evolved from dichotomous models (mastery goals and performance goals), through trichotomous models including goal approach–avoidance [25], 2 × 2 models with the achievement goal being the aim used to guide behaviour [26], to the 3 × 2 model [27]. This latter framework proposes three evaluation standards, task-based, self-based and others, which, together with the valence of competence, encompass six types of achievement goals [28]. In terms of sport, recent research emphasizes the role of trichotomous models and, specifically, the role of mastery goals (personal improvement) [29,30], establishing that perceived effort may be directly related to mastery achievement goals, fostering, in turn, sport commitment [31]. This model includes performance–approach goals (being better than the others) and performance–avoidance goals (avoiding being worse than others) [32].

In relation to Masters athletes and specifically Masters swimmers, we also failed to find studies that describe the different types of achievement goals under trichotomous models. Studies with other types of Masters athletes show that mastery goals have been positively related to enjoyment and personal well-being [33,34]. Performance–approach and performance–avoidance goals have been reported in previous studies with veteran athletes, and have also been shown to be highly unstable goals at the temporal level [21]. In this sense, it again seems important to establish a profile of achievement goals in Masters swimmers with the aim of knowing the types of goals that are predominant in such athletes.

### 1.4. The Theory of Sport Commitment

The theory of sport commitment defines the concept as the psychological state representing the resolve to continue participating in sport [35]. Since its initial formulation, the model has been expanded to include psychosocial aspects of sport engagement, such as enjoyment, personal investment, opportunities for involvement, social constraints, coach, family, etc., thus broadening its scope [16,36,37], including its association with achievement motivation and motivational climate [31,38]. Moreover, in veteran athletes, sports commitment has been related to aspects such as healthy lifestyles, enjoyment and wellness [39].

Adult-oriented sport practices have shown greater commitment, complementarity and coach closeness [40]. Many authors refer to the commitment of these athletes as “lifelong engagement in sport” [41,42]. In this sense, Orlick [43] proposes a model that considers not only current commitment, but adds future commitment measured through an athlete’s expectations of their sport. Athletes committed to sport throughout their lives might be expected to manifest this commitment equally in the present and in the future. However, this future commitment may be mediated by aspects that are, to a certain extent, beyond the control of the athletes, such as circumstances of family, work and health, among others. Several authors point to these psychosocial aspects as key motivators in how Masters athletes engage in their sport [44,45]. Determining the factors related to this future commitment may be important in promoting adherence to sport [46].

### 1.5. Aim of the Study

Drawing on these contributions, the present study is motivated by the scant body of research on the motivational characteristics of Masters swimmers, in general, and of Spanish Masters swimmers, in particular. The aim was to quantitatively analyse the predominant motivational regulation profile of Masters swimmers, the achievement goals that are most important to them and the profile of their current and future commitment to sport. Additionally, we sought to examine the possible relationships between these variables. In this way, we lay the foundations for future interventions and research lines with such athletes.

## 2. Materials and Methods

### 2.1. Design

The study design was empirical with descriptive and observational analysis, as the intention was to understand the motivation and sport commitment profile of the participants without manipulating the study variables [47].

### 2.2. Participants

The sample comprised 211 Masters swimmers (106 women and 105 men), aged between 20 and 74 years, with a mean age of 43.16 (SD = 13.09) (Table 1). The minimum number of participants required for the study was guaranteed (Section 3). The sample accounted for 29.68% of the total population under study. In addition, all the swimming clubs with a Masters section in the Autonomous Community of Castilla-La Mancha (21 clubs) were contacted and informed. Of these, 19 collaborated, with 90% of the clubs in the region thus being represented in the sample. This region is made up of five provinces, all of which were represented in the sample. Of the participants in this study, 20% had continued their swimming career with little interruption, 45% stopped swimming for more than two years (some as much as a decade) and 35% had not been swimmers (of these, 31% did other sports when they were young and 69% started swimming for various reasons). This sample profile is typical in “Masters or veterans” sports [48], with three types being identified, respectively, as “Rekindlers”, “Continuers” and “Late Bloomers”. Figure 1 present further demographic data on the sample.

### 2.3. Dependent, Covariate and Independent Variables

As dependent variables, we used the different sub-scales of the instruments administered (Section 2.3). As dependent variables of motivation regulation, we used intrinsic, identified, introjected and external regulation and amotivation (integrated regulation was not used as it is not included in the instruments administered, Section 3). Regarding achievement goals, the dependent variables were mastery goals, performance–approach goals and performance–avoidance goals. Finally, with respect to sport commitment, the dependent variables of current commitment and future commitment were used. Gender and age group were used as covariates, and as independent variables, the different groups that made up the sample were employed (overall sample, weekly training days, weekly training hours and engaging in other sports).

### 2.4. Instruments

For the purpose of the present research, the following instruments were used:Sport Motivation Scale (SMS) [49]. This questionnaire begins with the phrase “I practise sport…”, which is accompanied by 20 items that measure intrinsic, identified, introjected and external motivation and amotivation. These are assessed through 4 items, scored on a 5-point Likert scale, ranging from “strongly disagree” to “strongly agree”. The validation of the instrument in Spanish population yielded the following indices: *χ*^2^/*gl* = 4.804; *CFI* = 0.919; *TLI* = 0.904; *GFI* = 0.923; *SRMR* = 0.049 and *RMSEA* = 0.062. It is worth noting that in the model resulting from the validation process, intrinsic and integrated regulation were included in a single construct, which has also occurred when validating similar instruments among athletes in Spain [50].Achievement Goals Questionnaire for Sport (ACQS) [51]. This questionnaire begins with the phrase “In my sport…”, followed by 24 items scored on a 5-point Likert-type scale, where 1 is “strongly disagree and 5 is “strongly agree”. The aim of these items is to measure athletes’ orientation towards achievement goals across three sub-scales, namely, mastery goals, performance–approach goals and performance–avoidance goals. Its process of validation in Spanish population revealed a satisfactory fit to the data: *χ*^2^/*gl* = 2.2, *CFI* = 0.936, *SRMR* = 0.045, *RMSEA* = 0.048 (*IC* 90% = 0.042–0.054).Sport Commitment Questionnaire (SCQ) [46]. This questionnaire is introduced by the phrase “in my training sessions…”, which is followed by 11 items scored on a 5-point Likert-type scale, with responses ranging from “strongly disagree” to “strongly agree”. This instrument seeks to measure the level of athletes’ current and future commitment. Its validation in Spanish population yielded satisfactory indices: *χ*^2^ (23, *N* = 264) = 107.16, *p* = 0.005, *χ*^2^/*d*.*f*. = 2.49, *CFI* = 0.91, *IFI* = 0.91, *SRMR* = 0.06, *RMSEA* = 0.07.

To collect the data from the three questionnaires, an online application was created on the “Jotform” platform, which allowed the instruments to be accessed and completed on different devices in order to reach the maximum number of participants. In addition to the instruments described above, this application included questions on demographic data (category, gender, number and times of training sessions, etc.). Prior to all this, participants were informed of the aims of the research and all the pertinent permissions for participation in the research were requested.

### 2.5. Procedure

The study was proposed to the Swimming Federation of the Castilla-La Mancha region, of which the Masters were members. Following the approval of the Federation and its ethics committee, all the clubs with a Masters section in the region were contacted to provide them with the complete information required for the study. These clubs were tasked with transmitting this information to their Masters members, so that those who voluntarily wished to participate could do so. They were also given instructions on accessing the data collection application.

The clubs typically communicated with their athletes by means of a popular messaging application commonly used by the Masters to send and receive information about their sporting activity. The participants provided their data through the questionnaires used in a completely anonymous manner, and this information was administered and analysed only by the research team. The average time required to answer the questionnaire was approximately 10 min. All the participants voluntarily took part in this study after having been duly informed in advance of all aspects of the study and having signed the pertinent informed consent forms. Subsequently, once the information had been gathered, it was downloaded in .csv format and imported into statistical software applications. All the consent forms were stored in .pdf format.

### 2.6. Statistical Analysis

Prior to conducting the study, the necessary sample size was calculated using the G*Power 3.1 application (see Section 3). After data collection, we first calculated whether the sample followed a normal distribution (Kolmogorov–Smirnov) and whether it met the requirements of homoscedasticity (Levene) and sphericity (Mauchly).

We used IBM SPSS 28.0 for Mac to see whether the sample presented differences in the regulation of motivation (intrinsic, identified, introjected and external regulation and amotivation). To this end, we performed a mixed repeated measures analysis of variance (rANOVA), using, as repeated measures factors, the different sub-scales of the SMS, and, as covariates, the gender of the participants and age group. Additionally, as between-participants factors, we used the different groupings of the sample (overall sample, weekly training days, weekly training hours and practice of other sports).

To ascertain whether there existed differences in achievement motivation (mastery goals, performance–approach goals and performance–avoidance goals) within the sample, we ran a mixed repeated measures analysis of variance (rANOVA). The different sub-scales of the ACQS were used as repeated measures factors, the participants’ sex and age group were used as covariates and the abovementioned different groupings of the sample were used as between-participants factors.

To determine whether there were differences in sport commitment (present and future commitment) within the sample, we ran a further rANOVA. In this case, the different SCQ sub-scales were used as repeated measures factors, the participants’ sex and age group were used as covariates and the different groupings of the sample were used as between-participants factors.

After performing the rANOVAs, when necessary, post hoc results were calculated using the Bonferroni test. Correlations between the dependent variables were also estimated, using Pearson’s test. Finally, using the G*Power 3.1 program, we calculated the effect size of each test and the power of the effects obtained (error α 0.05). The criteria established to determine the power of the effect in the rAnova tests were low (≈0.01), medium (≈0.06) and high (≈0.14); in the post hoc results, the criteria were low (≈0.20), medium (≈0.50) and high (≈0.80) [52].

## 3. Results

### 3.1. Results of Calculating the Sample and Preliminary Analyses of Normality, Homogeneity and Sphericity

To ascertain the minimum sample size required for the study, the number of participants was calculated using the G*Power 3.1 application. The expected effect size (*f* = 0.25), the associated probability of error (*p* = 0.05) and the desired statistical power (*1 − β* = 0.80) were considered. The result obtained was a minimum sample of 120 participants. The results prior to the rANOVA tests showed that the sample followed a normal distribution in the variables used (type of motivation regulation, achievement goals and sport commitment) (1, 211) = 0.045, *p* > 0.200, (1, 211) = 0.035, *p* > 0.200, (1, 211) = 0.056, *p* = 0.60. Using Levene’s test, the homogeneity of the sample was tested in the variables of motivation regulation (1, 211) = 2.62, *p* = 0.107, achievement goals (1, 211) = 0.7, *p* = 0.405 and sport commitment (1, 211) = 1.10, *p* = 0.205. Finally, the Mauchly test was calculated to verify sphericity, obtaining significant results, and thus in the rANOVA calculations, the Greenhouse–Geisser correction was applied as it was considered the most conservative.

### 3.2. Multivariate Results of the rANOVAS

The multivariate tests yielded significant differences with very high effect sizes for type of motivation, *F* (1, 3.14) = 919.30, *p* < 0.001, *η*^2^ = 0.81, 1 − *β* = 0.93; achievement goals, *F* (1, 1.60) = 988.61, *p* < 0.001, *η*2 = 0.83, 1 − *β* = 1; and sport commitment, *F* (1) = 151.59, *p* < 0.001, *η*^2^ = 0. 41, 1 − *β* = 1. No significant differences were found between any of the dependent variables and the covariate variables and the rest of the groupings with their interactions (sex, age group, weekly training days, weekly training hours and practice of other sports), *p* > (0.05).

### 3.3. Results for Motivation Regulation

As regards motivation regulation, we found significant differences in each of the types of regulation, *p* < 0.001, with a very high effect size 1 − *β* > 0.80. We observed a predominance of intrinsic motivation and higher values in more self-determined motivations (Table 2). The motivation that scored highest was intrinsic motivation, followed within the self-determination continuum by the most self-determined motivations (identified and introjected). The lowest values corresponded to extrinsic motivation and demotivation.

### 3.4. Results for Achievement Goals

As regards the achievement goals, we found significant differences between the three variables, *p* < 0.001. The mastery goals were those most highly scored by the participants, obtaining significant differences with the performance–approach goals and performance–avoidance goals, with a very high effect size *1 − β* > 0.80 (Table 3). We also found significant differences between performance–approach goals and performance–avoidance goals. In this case, however, the effect size was medium, *1 − β* = 0.40.

### 3.5. Results for Sport Commitment

With respect to sport commitment, our aim was to see whether there existed significant differences between the current commitment reported by the participants and their future commitment. In this case, the participants’ current commitment was greater than their future commitment, *p* < 0.001, 1 − *β* > 0.80 (Table 4). No significant differences were found between any of the dependent variables and the covariate variables and the rest of the groupings with their interactions (sex, age group, weekly training days, weekly training hours and practice of other sports) *p* > (0.05)).

### 3.6. Correlations

The study variables were analysed using Pearson’s correlation test. The following significant correlations were found (Table 5).

## 4. Discussion

Based on the results obtained, we now proceed to describe the profiles related to the theories presented in the theoretical framework, analysed in terms of their predominant motivation regulations, their achievement goals and their current and future commitment to sport. Moreover, we discuss the relationships between the different study variables and how these results are associated with conclusions obtained in previous studies.

If we exclusively consider the offers of sports institutions, which are focused on developing competitive programmes for such athletes, it might be thought a priori that motivation would be fundamentally directed towards winning sports prizes, establishing performance–approach goals and a moderate commitment to achieving these outcomes. After analysing the swimmers’ motivation regulations, we observed significant differences between them all. Nonetheless, the motivation regulation perceived as less relevant for the Masters swimmers in this study was the external type, while the primary type of regulation was intrinsic motivation, although we should not forget that the measure used does not include integrated regulation, and thus a large part of this type of regulation would be associated with the intrinsic motivation sub-scale of the instrument [50].

This type of motivation is related to adherence to sports practice [53], the satisfaction of basic psychological needs [16,54], flow [55], enjoyment of the activity and physical and emotional well-being [11,48,56,57]. After intrinsic regulation, the next highest score corresponded to identified regulation, which is related to the benefits of the activity and which can have an impact on personal well-being [58]. Thus, we find that the autonomous types of regulation considered in the present study intercorrelated moderately (intrinsic–identified and integrated) and scored higher than the controlled types of regulation (introjected and external), which were also intercorrelated in a similar way to the former ones. These results are consistent with those obtained in other studies [49].

In our sample, it seems that aspects such as avoiding guilt or pride have a greater impact on motivation than the achievement of sporting rewards. Research has shown that, depending on the time of the season, athletes manifest different types of motivation regulations, with a number of authors even considering adjacent constructs [19,20] rather than a continuum. Meanwhile, the nature of the rewards in this type of activity is largely symbolic, with there being no significant financial rewards. Additionally, such rewards tend not to be valued by Masters athletes [41].

Concerning personal well-being, linked to the more self-determined types of regulation [11,48], various authors [59] find that Masters who begin or resume an activity primarily do so motivated by aspects related to well-being, more than simple sport performance. Additionally, Masters who rekindle their previous swimming careers recognize a change in their initial mentality based on sport performance, due to their interactions with other Masters who have been involved in the activity for longer [21,41,59]. In this sense, the fact that Masters show lower external regulation compared to the other types does not mean they do not value sport performance. Indeed, studies suggest they use competition as the reference for their training, while sports events themselves may contain important psychosocial elements [42]. In this regard, Young et al. [6], in a quantitative study, underlined that Masters swimmers in North America frequently framed their individual performances as a contribution to the team, something we also referenced in the introduction to this study. This implies that the overall result may be more significant than individual reward, as well as potentially generating social relationships. Thus, a distinction should be made between optimal and dysfunctional motivational variables at the level of the individual and of the social environment [60].

As regards achievement goals, our participants valued the intention to improve in their sport (mastery goals) over the intention to be better than others or to avoid being worse (performance–approach and performance–avoidance goals). We also found significant differences, although in this case moderate and with a low effect size, between approach and avoidance goals. These results suggest that the Masters participating in our study present positive adaptive processes related to their sport. Mastery goals have been negatively related to burnout and positively related to enjoyment, personal well-being, and relationship building [33,34]. In our study, mastery goals were moderately related to intrinsic motivation, but highly related to identified regulation. In other words, self-improvement-oriented goals are positively related to the benefit-oriented regulation generated by the activity.

Performance–approach and performance–avoidance goals are perceived as less important. Both goal orientations showed lower relationships with introjected regulation and medium-sized relationships with external regulation. These types of relationships have been reported in previous studies with veteran athletes, and have also been shown to be highly unstable goals at temporal level [21].

With respect to sport commitment, this study found that the Masters’ current commitment was significantly higher than future commitment. In previous studies, enjoyment of the activity, besides being the main source of intrinsic motivation, has been identified as the strongest determinant of sport commitment in veteran or Masters athletes [37,61]. In this sense, many authors refer to the commitment of these athletes as “lifelong engagement in sport” [41,42], and this commitment should thus manifest itself equally in the present and in the future. However, this future commitment may be mediated by aspects that are, to a certain extent, beyond the control of the Masters, such as circumstances of family, work, health, etc. Several authors point to these psychosocial aspects as key motivators in how Masters athletes engage in their sport [44,45]. In any event, the sport commitment of the participants in our study is considerably robust. Moreover, we found a strong relationship between current commitment, mastery goals and identified motivation. It seems logical that the Masters in our study are committed to their sport activity, because they seek their personal improvement through mastery goals and value the benefit the activity produces in this sense (identified regulation).

In short, our results show the Masters in our study present a motivational profile predominated by the most self-determined types of motivation (intrinsic, integrated and identified), mastery goals (personal improvement) and a strong current commitment to the sport. This motivational profile of self-determined motivation, mastery goals and high present commitment may be related to that reported in previous studies with veteran and Masters athletes [21]. However, some of the findings of the present study differ from those of other works, especially when Masters from other sports are analysed. In contrast to these previous studies, we found no age- or gender-related differences [21,41]. In other studies, men have presented higher achievement goals than women, while the latter showed higher intrinsic motivation scores. Age was a key factor related to personal well-being more than to sport outcomes. These results in our sample can be justified, on the one hand, by the gender parity of the participants, which is not so common in other sports, and, on the other hand, by the group effect in the Masters discussed above. This latter aspect generates a change in the perception of which goals are important, with a shift towards mastery goals and aspects linked to personal well-being and away from sports results, a finding also reported in other studies [21,59,61].

## 5. Conclusions

The aim of this study was to determine the predominant profiles of motivational regulations of Spanish Masters, their most valued achievement goals, and the level of their current and future sport commitment. We can conclude they present a motivational profile linked to the most self-determined types of regulation, intrinsic–integrated and identified regulation, high mastery goals and high current commitment to their sport practice.

The potential contribution of this study is that, to the best of our knowledge, aspects related to the development of motivational profiles have not previously been so extensively analysed in Masters swimmers and in Masters swimmers in Spain. Thus, we provide a base profile that may help researchers, coaches and sports institutions determine what motivates such athletes and so help improve and plan their sporting activity. Additionally, at the educational level, we address a phenomenon that involves “lifelong engagement in sport” [41,42], something that can be highly positive for individuals. Understanding the fundamental motivational aspects that generate such engagement can help educators, physical education teachers, coaches and trainers to shape pedagogical models in physical education and sports practice [7] in order to foster motivation and commitment, and, in turn, establish “lifelong engagement in sport” [41,42].

At the public health level, we provide valuable information on motivation. This can be used by institutions to promote motivation through interventions in sports programs in adult life, promoting adherence to sport and improving the general health of the participating population [7]. Finally, regarding sports policies, and answering the question proposed in Section 1.1 of this article, if the federations and institutions that govern sports policy take into account the motivational characteristics presented, they can intervene in different ways to promote lifelong engagement in sport [41,42,62]. On the one hand, they can take aspects into account when carrying out competitive events, and, on the other, they may implement interventions from an early age to develop motivational profiles that develop sports adherence [62].

As limitations of this work, which can be considered for future research, it would be highly recommendable to analyse, from the perspective of basic psychological needs (BPNs) [14], the role that might be played by the satisfaction of these needs (competence, autonomy and relatedness), and even that of the candidate basic psychological need of novelty [63,64,65]. Arguably, through the motivational profile found in our study, which encompasses satisfaction with practice, personal improvement, and autonomous types of regulation, the participants’ psychological needs of competence and autonomy may be satisfied. Moreover, the results of other studies underline the importance of psychosocial aspects [42]. We find that Masters swimming is a sport in which individuals contribute to the group, which suggests the basic psychological need of relatedness is satisfied. Thus, it seems relevant to examine whether the BPNs are satisfied in these athletes.

## Figures and Tables

**Figure 1 behavsci-13-00828-f001:**
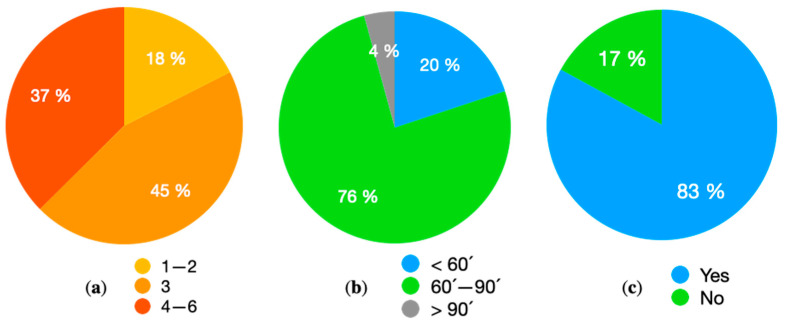
(**a**) Days of weekly training; (**b**) minutes of daily training; (**c**) practice another sport.

**Table 1 behavsci-13-00828-t001:** Number and percentage of participants by Masters age groups.

Age Groups	*n*	%	% Accumulated
+20	11	5.2	5.2
+25	19	9.0	14.2
+30	24	11.4	25.6
+35	17	8.1	33.6
+40	33	15.6	49.3
+45	48	22.7	72.0
+50	31	14.7	86.7
+55	13	6.2	92.9
+60	10	4.7	97.6
+65	4	1.9	99.5
+70	1	0.5	100.0
Total	211	100.0	

**Table 2 behavsci-13-00828-t002:** Results of the post hoc tests on the type of motivation variable.

Motivation Regulation	M	SD	*p*	(1 − *β* Err Prob)
Intrinsic–Identified	4.58	0.51	<0.001 *	0.99
	4.17	0.76		
Intrinsic–Introjected	4.58	0.51	<0.001 *	1
	2.92	1.11		
Intrinsic–Extrinsic	4.58	0.51	<0.001 *	1
	1.80	0.89		
Intrinsic–Amotivation	4.58	0.51	<0.001 *	1
	1.15	0.36		
Identified–Introjected	4.17	0.76	<0.001 *	1
	2.92	1.11		
Identified–Extrinsic	4.17	0.76	<0.001 *	1
	1.80	0.89		
Identified–Amotivation	4.17	0.76	<0.001 *	1
	1.15	0.36		
Introjected–Extrinsic	2.92	1.11	<0.001 *	1
	1.80	0.89		
Introjected–Amotivation	2.92	1.11	<0.001 *	1
	1.15	0.36		
Extrinsic–Amotivation	1.80	0.89	<0.001 *	1
	1.15	0.36		

Note: M = mean, SD = standard deviation, * (*p* < 0.001), (1 − *β* err prob) = effect power.

**Table 3 behavsci-13-00828-t003:** Results of the post hoc tests on the achievement goal variables.

Achievement Goals	M	SD	*p*	(1 − *β* Err Prob)
MGs–PAGs	4.62	0.50	<0.001 *	1
	2.21	0.98		
MGs–PAvGs	4.62	0.50	<0.001 *	1
	2.07	0.93		
PAGs–PAvGs	2.21	0.98	<0.01 **	0.40
	2.07	0.93		

Note: MGs = mastery goals, PAGs = performance–approach goals, PAvGs = performance–avoidance goals, M = mean, SD = standard deviation, * (*p* < 0.001), ** (*p* < 0.01), (1 − *β* err prob) = effect power.

**Table 4 behavsci-13-00828-t004:** Results of sport commitment.

Sport Commitment	M	SD	*p*	(1 − *β* Err Prob)
CC–FC	4.20	0.38	<0.001 *	1
	3.72	0.53		

Note: CC = current commitment, FC = future commitment, M = mean, SD = standard deviation, * (*p* < 0.001), (1 − *β* err prob) = effect power.

**Table 5 behavsci-13-00828-t005:** Significant correlations between the dependent variables.

Variables	R^2^	(1 − *β* Err Prob)
Intrinsic–Identified	0.401	1
Intrinsic–Amotivation	0.315 *	0.99
Intrinsic–Mastery Goals	0.252 *	0.96
Intrinsic–Current commitment	0.252 *	0.96
Intrinsic–Future commitment	0.201 **	0.83
Identified–Introjected	0.248 *	0.95
Identified–Mastery goals	0.551 *	1
Identified–Current commitment	0.481 *	1
Identified–Future commitment	0.348 *	0.99
Introjected–External	0.459 *	1
Introjected–Perf.-Appr. Goals	0.218 **	0.89
Introjected–Perf-Avoid. Goals	0.264 *	0.97
External–Amotivation	0.192 ***	0.80
External–Perf.-Appr. Goals	0.460 *	1
External–Perf-Avoid. Goals	0.454 *	1
Amotivation–Perf.-Appr. Goals	0.180 **	0.74
Amotivation–Perf-Avoid. Goals	0.222 **	0.90
Mastery goals–Current commitment	0.630 *	1
Mastery goals–Future commitment	0.464 *	1
Perf.-Appr. Goals–Perf-Avoid. Goals	0.758 *	1
Perf.-Appr. Goals–Current commitment	0.143 ***	0.54
Perf.-Appr. Goals–Future commitment	0.248 *	0.95
Perf-Avoid. Goals–Future commitment	0.159 ***	0.63
Current commitment–Future commitment	0.684 *	1

Note: * (*p* < 0.001), ** (*p* < 0.01), *** (*p* < 0.05), (1 − *β* err prob) = effect power.

## Data Availability

The data presented in this study are available on request from the corresponding author. The data are not publicly available due to privacy.

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
