# Peer review of "Self-Determined Regulation, Achievement Goals and Sport Commitment in Spanish Masters Swimmers"

_behavsci, 2023, doi:10.3390/bs13100828_

Round 1
Reviewer 1 Report
Dear Authors,
"Self-determined Regulation, Achievement Goals and Sport Commitment in Spanish Masters Swimmers".
The study addresses an interesting and relevant topic, it is well written, well structured and clear. The methodology is adequate to the objectives and can be replicated to analyze the motivational characteristics of athletes in other sports. The results are well presented with coherent conclusions. However, I suggest to reduce the number of keywords.
Kind regards
Author Response
Dear reviewer. Thank you very much for your review, which helps us improve our paper.
Attached we send you a document with the requests you have made to us. We believe that we have correctly resolved them.
All changes requested by all reviewers are in the paper highlighted in red.
Thank you very much again.

Reviewer 2 Report
This is a well written and interesting article. Very easy to follow and read making it relevant for all levels of scholars. A bit of work is needed in providing a couple more recent articles for the literature and elaborate a bit more on the contribution of this work mostly from a practical point of view.
Well written and interesting easy to follow and read article, making it relevant for all levels of scholars.
Author Response

(The authors gave the same response as above.)

Reviewer 3 Report
Dear authors,
About your paper titled: Self-determined Regulation, Achievement Goals and Sport Commitment in Spanish Masters Swimmers.
First, thank you for the opportunity to review this work. I would like to provide some suggestions:
Keywords: Please avoid using "MASTERS" too many times in the section. To me, one time is sufficient.
Introduction:
During your introduction, you should cite recent articles about the relevant topic. My major concern is "What are academic contributions of the current study? To me, the authors simply took some popular concepts from the previous studies and measured in Spanish context.
The authors should provide improved literature review to justify the importance of this study rather than to just investigate those concepts in the specific context.
Discussion:
Please work on practical implications from the findings of this study.
NA
Author Response

(The authors gave the same response as above.)

Reviewer 4 Report
Overall, this was a good attempt. However, please address the following issues/questions:
- Re-state the study aims for clarity.
- in a separate section, isolate and explain each theoretical framework...and then in your discussion of results, explain how you utilized the stated theories.
-check on spelling. There are multiple typos throughout the manuscript (See section 2.1 (desing' instead of 'design' for example.
-expand Section 5 - 'Conclusion' to include future studies, study gaps, and/or the potential impacts of the study (pertaining to policy, theory, and training, etc.)
See above comments.
Author Response

(The authors gave the same response as above.)

Round 2
Reviewer 3 Report
I am fine with this revision.